# Economic burden and catastrophic cost among people living with sickle cell disease, attending a tertiary health institution in south-east zone, Nigeria

C. N. Amarachukwu [1]*, I. L. Okoronkwo[1], M. C. Nweke[2], M. K. Ukwuoma[3]

1 Department of Health Administration and Management, University of Nigeria, Enugu Campus, Enugu, Nigeria, 2 Department of Physiotherapy, School of Basic Medical Sciences, College of Medicine, University of Benin, Benin City, Nigeria, 3 Department of Medical Rehabilitation, University of Nigeria, Enugu Campus, Enugu, Nigeria

☯ These authors contributed equally to this work.
* charityamarachukwu03@gmail.com

**Data Availability Statement:** The data are available at: https://figshare.com/s/5a8da1354294b65da390

## Abstract

Out-of-pocket spending and lack of adequate health policy support for people living with sickle cell disease in Nigeria may predispose to high economic burden and catastrophic cost. The objective of the study was to evaluate the economic burden and catastrophic cost of sickle cell disease patients in a Nigerian tertiary health institution. In this study, a cross-sectional descriptive survey design was used to study a sample of 149 sickle cell disease patients managed at University of Nigeria Teaching hospital Enugu, South east Nigeria. A structured pre-tested interviewer-administered questionnaire was used to collect primary data from adult participants and caregivers of paediatric patients. Data collection lasted for three months. The major findings were median monthly economic burden of approximately N76, 711 (US$385) per person. Of this, outpatient cost constituted approximately 88%. Admission, drugs and blood transfusion constitute the major contributors to the economic burden experienced by the sickle cell disease patients in the study. All socio-economic status groups suffered catastrophic expenditure but the poorest quartile had the highest incidence: 61% at 40% threshold, 71% at 30% threshold and at 88% at 10% threshold. **Conclusion:** economic burden was high for sickle cell disease patients who also suffered high catastrophic costs due to the impact of out-of-pocket expenditure. People living with sickle cell disease need financial protection especially for the poorest since they buy from the same market and incur same costs. Policy decision making to assist the sickle cell disease patients cope with cost of care is needful in Nigeria.

## Introduction

Sick cell disease (SCD) is a major public health problem in Africa [1]. The estimated number of worldwide annual birth of people with SCD is about 250,000 [2], and Africa represents about 85% of global burden of SCD with about 200,000 to 230,000 [1, 3, 4]. There is a wide

and https://doi.org/10.6084/m9.figshare.20364432.

**Funding:** The authors received no specific funding for this work.

**Competing interests:** The authors have declared that no competing interests exist.

variation in the prevalence of the gene in different parts of Africa. However, the frequency of the trait has been estimated as high as 10–50% [4–6]. The most vulnerable age group is the under-five and accounts for about two-thirds of deaths in children fewer than 5 years. In Africa, SCD accounts for more than 6% of all deaths in African children younger than five years [7], and most of the infant deaths due to SCD in Africa occur in West African countries with Nigeria alone accounting for about 80% of the deaths [7, 8]. In Nigeria, SCD remains the most frequent and traumatizing genetic disease which continues to devastate the families of sickle cell patients both mentally and economically. Nigeria remains a home for the largest Africa sickle cell disease patients [9]. It is estimated that approximately 2.3% (150,000 children) are born with sickle cell disease each year [9], and about 5% of them die before they reach age five [10].

There is no universal cure of SCD in the world [11]. Although developed countries such as USA and UK are today making successful attempts to cure sickle cell disease using bone marrow transplant or stem cell transplant, such technologies are expensive for resource poor countries [5], including Nigeria, and patients are unable to find well-matched donors thus discouraging its use for most patients [12]. Hitherto, the treatment of SCD is geared towards limiting the frequency or number of crisis, preventing infection, reducing organ damage and minimizing pain and discomfort. The chronic nature of SCD associates it with high healthcare utilization and can cause a significant economic loss and pose as burden on households and individuals [6]. It affects them through direct long-term out of pocket expenditures for treatment and diagnosis of the chronic nature of sickle cell complications [13], and the indirect costs resulting from productivity losses due to patient disability and premature mortality, time spent by family members accompanying patients when seeking care, and intangible costs such as psychological pain to the family and loved ones. SCD exposes the affected individuals and their families to catastrophic health expenditures especially those individuals from poor and marginalized families [6].

Catastrophic Healthcare expenditure is very high healthcare spending beyond which individuals begin to sacrifice consumption of basic needs. It is equal to or in excess of 40% of non-subsistence income consumption [14], that is income available after basic needs have been met (non-food expenditure), although countries could set their thresholds based on their peculiarities [15]. SCD patients in Nigeria experience catastrophic expenditure not only because they visit the health facilities frequently to treat the associated complications but most times present late with complications often associated with poverty. Previous studies peg catastrophic level at 5–40% [14, 16–18], however, it is important to contextualize the Nigerian situation where more than 70% of the population live below $1 per day and citizens spend more than 40% of their total income to satisfy hunger [19]. Private funding constitute more than 90% and prepayment risk pooling mechanisms is lacking [17, 18, 20]. Out-of pocket spending (OOPS) is the predominant payment mechanism for healthcare in Nigeria and it is associated with ever increasing healthcare cost and difficulty in estimation of economic impact of healthcare expenditure on individuals challenged with illness [21].

Apart from the time and financial costs faced on travelling and waiting for medical care services at the clinics, the patients and or their families may find themselves facing enormous costs if they still have to pay for the medical services at the point of care. Increased out of pocket health expenditures can lead to crowding out of other essential consumption items such as food, housing and education [22]. Furthermore, adult persons living with SCD may find themselves in a situation whereby they have to sacrifice paying medical costs or other household needs like food and education for their children to meet the costs of their own suffering. Such patients may also lose time for participation in economic activities during the period in which they are responding to illness episodes [11, 23, 24]. Excessive reliance on

OOPS exacerbates the already inequitable access to quality care and exposes households to the financial risks of expensive illnesses like SCD [25].

It is worrisome that despite the fact that UN in 2011 elated the status of non-communicable diseases (NCDs) to that of Human immunodeficiency virus infection, TB and Malaria because of their economic and health importance [16, 26], no support or financial risk protection exist for SCD which is assuming an increasing proportion and affects vulnerable age group-young people [7]. Obviously, there is need for more presence of development partners and non-governmental organization (NGOs) in the fight against SCD and inherent economic burden especially in Nigeria. Perhaps the reason for the negligence may be due to paucity of data demonstrating in quantitative terms the economic burden and catastrophic cost of SCD in Nigeria. Evidence-based data is required to move SCD into health policy agenda in Nigeria. Hence the study aimed to evaluate economic burden and catastrophic cost of SCD patients attending adult and paediatric haematology outpatient clinic in the University Teaching Hospital Ituku-Ozalla, Enugu.

## Materials and methods

### Study design and setting

This study was a cross-sectional descriptive study among sickle cell patients attending Adult and Paediatric Haematology Outpatient clinic in the University Teaching Hospital Ituku-Ozalla Enugu. This was deemed fit because "cost-of-illness" estimate represents a descriptive economic method which is often used to estimate cost of a particular disease [27]. Cross sectional descriptive survey design was therefore considered appropriate because the purpose of this study is to observe, describe, estimate and document the cost of SCD.

The study was conducted at the Paediatric and Adult Haematology Outpatient Clinic of the University of Nigeria Teaching Hospital (UNTH), Ituku Ozalla, in Enugu state, South-east Nigeria. UNTH is a Federal Government-owned tertiary health facility and the pioneer teaching hospital located at Ituku-Ozalla in Enugu State, a semi urban centre on the outskirts of Enugu city along the Enugu-Port Harcourt express way, about 21km from Enugu City covering approximately 306 hectares of land. The hospital was established by decree number 23 of 1974 of the Federal Military Government. It's a 500-bed tertiary hospital. The hospital serves both self and health system referred patients from the seventeen Local Government Areas of Enugu State as well as some parts of its neighbouring South Eastern States- Imo, Abia, Anambra and Ebonyi. The hospital is the main referral facility providing both adult and paediatric sickle cell care for many communities in Enugu and surrounding south-eastern States. The study was approved by the Ethics Research Committee of University of Nigeria Teaching Hospital Ituku-Ozalla, Enugu (NHREC/05/01/2008B-FWA00002458-1RB00002323).

### Study participants

Participants in this study constituted all patients who attended the adult and paediatric sickle cell outpatient clinic from November 2015 to January 2016. About 480 SCD patients attended the children clinic and about 400 patients attended the adult clinic in the last one year (September 2014 to October 2015). Hence, the estimated target population was 880 patients. Eligibility criteria were individuals receiving treatment in the centre for the past 1 year (November, 2015- January, 2016) and who were actively involved in the management of the condition during the period of data collection, well informed about the cost of care and willing to participate in the study. For dependents/children, consent was obtained from their parents or guardian, and all questions were directed to them.

## Variable of interest

In this study, variables of interest comprised economic burden, catastrophic diabetic cost, and socioeconomic status. Economic burden was approximated by unit cost of all SCD out-patient services received. Catastrophic diabetic cost was measured by non-food consumption expenditure of the respondents (i.e. income) plus the direct cost of SCD. Socioeconomic status group was valued by number of household items owned by respondents.

## Sample

A minimum sample size of 140 was calculated using the formula, $n = n_0/1 + (n_0 − 1)/N$, after estimating a single finite proportion from a target population of 880 physician diagnosed SCD patients (adults and children). It was anticipated that as many as 10% might withdraw from the study prior to its completion through possible refusal to continue with the study. Thus, the formula $q = n/1 − f$ where q is the adjustment factor and f is the estimated non response rate was used [28, 29]. 10% of the minimum sample size was then added. However, 149 participants completed the questionnaire. Consecutive sampling technique was used in the study.

## Data collection

In this study, a pre-tested semi-structure questionnaire was used to collect primary data from adult participants and caregivers of paediatric participants. The questionnaire comprises four sections, but three sections were utilized and reported in this study namely sections A, B, and D. Section A consisted of demographic and socio-demographic characteristics of the respondents and their caregiver's where applicable. Section B dwelt on the economic burden of sickle cell disease patients (direct costs of accessing SCD care and indirect cost of earnings lost as a result of time spent visiting healthcare system and being absent from work). Regarding indirect cost, the monetary value of time loss was computed using 21 days a month for public servants and daily wages for self-employed and housewife. Time lost due to SCD care was translated into money by using the man- hour earnings based on employment status to get the earnings of respondents/caregiver. Total cash income lost by respondent/caregiver was valued by multiplying the number of days absent from work by daily man-hour based on employment status multiplied by earnings from income sources.

Section D assessed socio-economic status and catastrophic SCD cost. The questionnaire was validated using face and content validity. The instrument was submitted to experts in health economics and a consultant Haematologist who assessed it for face and content validity. Their inputs were used to effect corrections and the instrument used as a valid instrument for data collection.

The questionnaire was pilot tested on 7 patients attending sickle cell clinic in Enugu State University Teaching hospital Parklane using split method. This low number of respondents was because of unavailability of patients in the hospital. The responses of aim were subjected to internal consistency test using coefficient alpha (Cronchach's alpha) method. The co-efficient of reliability obtained by sections A, B and D were 0.23, 0.55, and 0.5 respectively. The low reliability coefficient 0.23 obtained for section D was due to heterogeneity created by the gender. The study was not directed at ascertaining the influence of socio-demographic characteristics on economic cost or catastrophic cost, therefore, the low coefficient was considered inconsequential for the study outcome. Participants were informed that participation is voluntary. The principal investigator and two research assistants trained on the purpose of the study and how to administer the instrument collected the data were involved. The instrument was administered within the hours of 9 am and 3pm before or after seeing their physicians.

## Data analysis

Data gathered were collated, tallied, grouped and analysed using Statistical Package for Social Sciences (SPSS) version 15.0. Data on socio-demographic characteristics were presented using descriptive statistics of frequencies, percentages means and standard deviations were presented. Economic burden (direct cost and indirect costs) were derived using descriptive statistics. The Kruskal–Wallis non-parametric test (which reports a $\chi^2$ statistic) used to compare the means of health expenditures. The monetary value of man hour lost was calculated using the Human Capital Approach (descriptive economic method). The monetary value of time loss was computed using 21 days a month for public servants and daily wages for self-employed and housewives. Time lost due to SCD care was translated into money by using the man- hour earnings based on employment status to get the earnings of respondents/caregivers. Socio-economic status was determined using principal component analysis (PCA) in STATA Software STATA V.11 version (StataCorp LP, USA). PCA allows to convert series of ownership variables into socio-economic status. The first component of the PCA was used to derive weight to form an assets- based socio-economic index which was used to categorize the respondents into four socioeconomic quartiles (q1-q4) of poorest, poorer, poor and least poor [30]. Measure of in-equality was the ratio of the mean of the poorest SES group over that of the least poor. Catastrophic SCD cost was determined as a proportion of SCD cost and non-food expenditure. Catastrophe was checked at fixed threshold of 40%. The association between socio-economic status and Catastrophic SCD costs was assessed using Chi square statistics. Alpha was set at 0.05.

## Results

Of the 156 questionnaires administered, 149 were correctly filled and completed and data were analysed and presented in Tables. Table 1 showed that more than half of the patients were males and single. Approximately half (49.7%) of the patients were young adults, 73 (46.2%) were children (toddlers (7.4%), school age (23.5%) and teens (18.1%)). More than half 92 (61.7%) of the patients have had SCD for one to ten years. On the frequency of check-up, majority of the patients fell within the category of once per month check-up 93(62.4%) followed by twice a month check-up 32(21.5%). Almost all (99.3%) of the patients had formal education, with primary education numbering highest (48%). Most (69.7%) of the respondents were schooling and a few(14.1%) were employed.

Table 2 showed the direct (in-patients and out-patients) cost of treating SCD. The median total direct cost per patient per month wasUS$385 (187.3–697.1), with in-patient direct cost constituting the bulk-US$338.61 (168.8–661.2). Cost of admission ranked highest (US$133.22 (59.60–214.56)), followed by drugs (US$62.74 (20.08–99.52)) and transfusion, US$59.72 (36.51–101.63). Twenty eight respondents were admitted within the months of study. Of the inpatient cost, cost of admission ranked highest (US$133.22 (59.60–214.56)), followed by drugs (US$62.74 (20.08–99.52)), and transfusion (US$59.72 (36.51–101.63)). Physiotherapy (US$20.58 (4.14–57.59)), drugs (US$15.06 (9.18–18.79)) and laboratory investigations (US$10.54 (5.02–19.07)) were the most important source of outpatient direct cost.

Table 3 showed the indirect (in-patients and out-patient) cost of treating SCD. The average number of caregivers that stayed with a patient on admission was 2.03±1.94. Average number of days spent in the hospital on admission was 5.54±4.62days. Total cash in-patient-related loss by respondents/patients was US$43.30±5.67. Total cash loss by respondents was 63.04 ±19.07, with total out-patient loss constituting US$19.73±13.40.

Table 4 displays respondents categorized into four socioeconomic status groups using principle component analysis (PCA) on STATA software to generate an assets based index. The

**Table 1. Selected socio-demographic characteristics of the patients and respondents.**

| Variables | N(%) |
|---|---|
| Patients' gender | |
| Male | 96 (64.4) |
| Female | 53 (35.6) |
| Patients' age (years) | |
| Toddlers (1-4years) | 11 (7.4) |
| School age (5-11years) | 35 (23.5) |
| Teens (12-17years) | 27 (18.1) |
| Young adults (18-39years) | 74 (49.7) |
| Middle age (40-54years) | 1 (0.7) |
| Patients' marital status | |
| Single | 131 (87.9) |
| Married | 18 (12.1) |
| Divorce | - (-) |
| Time since SCD diagnosis | |
| 6months after birth | 26 (17.4) |
| 6 to <12months | 12 (8.1) |
| 1 to <5years | 54 (36.2) |
| 5 to 10years | 31 (20.8) |
| >10years | 20 (13.4) |
| Others | 6 (4.0) |
| No of times patients needed care in the past 1 month | |
| Once | 93 (62.4) |
| Twice | 32 (21.5) |
| Thrice | 15 (10.1) |
| Four times | 2 (1.3) |
| Five times | 4 (2.7) |
| Six times | 2 (1.3) |
| Patients' employment status | |
| Unemployed | 24 (16.1) |
| Civil service | 6 (4.0) |
| Private sector | 6 (4.0) |
| Self employed | 8 (5.4) |
| Housewife | 1 (0.7) |
| Schooling | 104 (69.8) |
| Respondents' education level | |
| No formal education | 2 (1.3) |
| Primary education | 48 (32.2) |
| Junior secondary | 16 (10.7) |
| Secondary | 40 (26.8) |
| University/college/polytechnic | 40 (26.8) |
| Postgraduate | 3 (2.0) |

highest weight was assigned to radio (0.589), followed by bicycle (0.486) and fan (0.381) and so on. We identified four socioeconomic statuses. Respondent of poorest socio-economic status numbered highest (27.5) followed by those (24.8) of least poor socio-economic status.

Table 5 shows that at 40% threshold, the levels of catastrophic cost were 61%, 36%, 46% and 32% for q1, q2, q3 and q4 respectively. At 30%, the levels were 70.7%, 50.0%, 57.1% and 40.5%,

**Table 2. Direct cost of SCD per month reflecting unit cost.**

| Cost | IPD Median (IQR) | OPD Median (IQR) | Total Median (IQR) |
|---|---|---|---|
| **Direct Medical Cost (US$)[a]** | | | |
| Registration | 6.52 (3.26–9.53) | 3.26 (3.26–5.96) | 24.09 (6.52-na) |
| Consultation | 3.39 (3.01–6.78) | 3.01 (3.01–3.26) | 8.78 (6.52–9.84) |
| Laboratory | 15.06 (12.80–36.89) | 10.54 (5.02–19.07) | 33.63 (26.25–112.42) |
| Admission | 133.22 (59.60–214.56) | - | 133.22 (59.60–214.56) |
| Drugs | 62.74 (20.08–99.52) | 15.06 (9.18–18.79) | 67.25 (27.55–112.42) |
| Blood transfusion | 59.72 (36.51–101.63) | - | 59.72 (36.51–101.63) |
| Physiotherapy | 15.06 (na) | 20.58 (4.14–57.59) | 16.06 (5.02–50.19) |
| Diet | 20.83 (7.28–44.54) | 5.02 (3.51-na) | 14.05 (5.02–27.60) |
| **Direct non-medical cost (US$)[a]** | | | |
| Transportation | 5.02 (2.76–11.04) | 2.46 (1.47–4.02) | 8.28 (5.2–19.07) |
| Disposable | 19.61 (8.46–147.79) | 7.53 (2.26–30.34) | 14.55 (6.58–31.81) |
| Other expenditure | 15.06 (15.06-na) | 4.68 (2.01–10.03) | 4.68 (2.01–10.03) |
| Monthly cost per patient | 338.61 (168.8–661.2) | 72.14 (33.9–149.1) | 385 (187.3–697.1) |

[a]: **1US$ = N199.25;** IPD: in-patient department; OPD: out-patient department

32% respectively, while at 10% threshold, level of catastrophic cost were 87.3%, 83.3%, 85.7% and 64.9% for qi, q2, q3 and q4 respectively. Majority that suffered catastrophic expenditure at 40% threshold belong to the poorest SES group 25(61.0%). Only 12 (32.4%) of the least poor SES group experienced financial catastrophe at this level. At 30% threshold catastrophic spending was highest (70.7%) among the poorest followed by the poor 20(57.5%). All SES quartile suffered catastrophic expenditure but the poorest quartile had the highest incidence: 61% at 40% threshold, 71% at 30% threshold and at 88% at 10% threshold. The mean non-food expenditure for qi, q2, q3 and q4 were US$193.79, US$250.92, US$33 and US$525.94 respectively. The ratio of q1/q4 was 1:3 meaning that non-food expenditure of the least poor was approximately 3 times that of the poorest group.

The table showed that at 40% and 30% threshold, there was no significant association between types of expenditure and socioeconomic status (P> 0.05 in each), however, at 10%, a significant association was found between types of expenditure and socio-economic status (P = 0.047).

## Discussion

The median monthly in-patient direct cost of N67468 (US$338.6) recorded in this study is considered rather high for respondents. This is inconsistent with the work of Adegoke et al. [7], conducted in South-western Nigeria, in which lower cost was reported. This high cost may be due to difference in location, with South-eastern Nigeria having higher cost of living than

**Table 3. Indirect cost of SCD per month reflecting unit costs.**

| Indirect Cost of Treating Sickle Cell (US$)[a] | IPD Mean (SD) | OPD Mean (SD) | Total Mean (SD) |
|---|---|---|---|
| No of care givers at the facility | 2.03 (1.94) | 0.9 (1.11) | 2.93 (3.05) |
| Total number of days absent from work | 5.54 (4.62) | 1.39 (0.82) | 6.93 (5.44) |
| Total cash income lost by respondent | 43.30 (5.67) | 19.73(13.40) | 63.04 (19.07) |

[a]: **1US$ = N199.25;** IPD: in-patient department; OPD: out-patient department

**Table 4. Socioeconomic status as represented by household items owned by respondents on assets based index (n = 149).**

| Household item | Weight | Yes | No |
|---|---|---|---|
| Radio | 0.589 | 116 (77.9%) | 33 (22.1%) |
| Fridge | 0.175 | 130(87.2%) | 16 (12.8%) |
| Television | 0.231 | 143 (96.0%) | 6 (4.0%) |
| Fan | 0.381 | 136 (91.3%) | 13 (8.7%) |
| Air conditioner | 0.116 | 28 (18.8%) | 121 (81.2%) |
| Personal Computer | 0.129 | 70 (47.0%) | 79 (53.0%) |
| Bicycle | 0.486 | 25 (16.8%) | 124 (83.2%) |
| Motorcycle | 0.241 | 11 (7.4%) | 138 (92.6%) |
| Tricycle (Keke) | 0.220 | 10 (6.7%) | 139 (93.3%) |
| Motorcar | 0.212 | 73 (49.0%) | 76 (51.0%) |
| Kerosene lamp | 0.100 | 71 (47.7%) | 78 (52.3%) |
| Generator | 0.171 | 110 (73.8%) | 39 (26.2%) |
| Rechargeable lamp | -0.111 | 117 (78.5%) | 32 (21.5) |
| Gas cooker | 0.195 | 67 (45.0%) | 82 (55.0%) |
| Stove | 0.142 | 130 (87.2) | 19 (12.8%) |
| Microwave | -0.137 | 41 (27.5%) | 108 (72.5%) |
| Washing Machine | -0.343 | 25 (16.8%) | 124 (83.2%) |
| **Socio-economic status** | **Quartiles** | **Frequency** | **Percentage** |
| The poorest | Q1 | 41 | 27.5% |
| The very poor | Q2 | 36 | 24.2% |
| The poor | Q3 | 35 | 23.5% |
| The least poor | Q4 | 37 | 24.8% |

South-western Nigeria [31]. However, it is not be unrelated to the number of admission, frequent visits to hospital due to acute episodes and intensity of care, with an average of 5 days on each admission in our study. The high proportion of SCD-related costs associated with inpatient hospitalizations suggests that interventions that reduce complications such as pain crises and anaemia and require hospitalization could be expensive but cost-effective. The out-patient cost of treating SCD patient was US$72.14 (33.9–149.1). This is lower than the cost obtained in

**Table 5. Catastrophic expenditure among various socio-economic groups of SCD households.**

| Expenditure | poorest n = 41 | very poor n = 36 | poor n = 35 | least poor n = 37 | p-value |
|---|---|---|---|---|---|
| Non-food Expenditure | US$193.79 | US$ 250.92 | US$332 | US$525.94 | |
| Ration of non-food $q^n/q^1$ | 1 | 1.29 | 1.71 | 2.71 | |
| Ratio $q^n/q_4$ | 0.37 | 0.48 | 0.63 | 1 | |
| Catastrophic threshold | | | | | |
| **10%** | | | | | |
| Not catastrophic | 5 (12.2) | 6 (16.7) | 5 (14.3) | 13 (35.1) | 7.953 |
| Catastrophic | 26 (87.8) | 30 (83.3) | 30 (85.7) | 24 (64.9) | 0.047 |
| **30%** | | | | | |
| Not catastrophic | 12 (29.3) | 18 (50.0) | 15 (42.9) | 22(59.5) | 7.655 |
| Catastrophic | 29(70.7) | 18 (50.0) | 20(57.1) | 15(40.5) | (0.054 |
| **40%** | | | | | |
| Not catastrophic | 16(39.0) | 23(63.9) | 19 (54.3) | 25(67.6) | 7.739 |
| Catastrophic | 25(61.0) | 13(36.1) | 16(45.7) | 12(32.4) | (0.052) |

the work of Amendah et al. [32], in which out-patient cost ranged from US$94 to US$229. The discrepancy in cost estimate could be due to difference in population studied as well as difference in cost of living. The Kenyan study was restricted to children unlike our study which involved both children and adults; however, the higher cost found in our study is strange as unit cost of an adult medication is expected to be higher than that of a child. Hence, the discrepancy may be due to difference in cost of living than population studied. Nonetheless, both studies have highlighted the high magnitude of the economic burden of SCD and cost implication which tends to increase as the population advances in age. This is well elaborated by Kauf et al. [33], in a study which demonstrated that SCD cost increased as patients advance in age. Most respondents visited the clinic once per month, hence the cost is adjudged to be high in a country where 70% of the population live in poverty and earn less than one US dollar per day and spend more than 40% of their income to satisfy hunger [19]. More worrisome is the fact that healthcare delivered in public facilities in Nigeria is not subsidized for patients amid poor health insurance coverage.

The cost of admission deposits ranked highest, drugs and blood transfusion in the in-patients. The fact that blood transfusion ranked top three affirms the fact that SCD is a blood disorder and depletes the red blood cells. Hence, blood transfusion necessitated by anaemic crisis is a frequent form of management for SCD especially when on admission [34]. In out-patient, physiotherapy and drugs are the major components of direct cost. This disagrees with the study by Adegoke et al. [7], where the major components of cost of hospitalizations by children with SCD were investigations followed by drugs. The difference could be due to the application of the statistic of median instead of mean. In addition to confirming the high cost of direct cost of SCD, our results indicate that patients with SCD incur substantial indirect costs as well. The indirect in-patient cost of US$63.04±19.07 were considered high. The accompanied caregivers sacrifice their business time, their work hours in the course of taking care of a sick person or in the course of receiving care themselves in the hospital. They suffer some economic loss any day they accompany the patient, or the adult patient loses income because he has gone to the hospital. The high indirect cost of SCD observed in this study may be related to complications and the fact that the majority (71.3%) of respondents in the present study were employed, 34% being self-employed (most trading). The respondents spent 5.4days a month on admission (in-patient) with an average of 2 caregivers, and lost 4hours attending out-patient check-ups at a check-up appointment rate of 1.4days per month. This implies that respondents especially the self-employed lost much of their monthly earnings seeking care. The finding from this study is in agreement with the study of Adegoke & Kuteyi [9], stating that poor health status of children with SCD reduces caregiver's employability and worsen the socio-economic burden of families. Moskowitz et al. [35], in a study stated that up to 24.3% of caregivers in the USA missed two or more days of work per 3days hospital admissions of their children with SCD.

The treatment cost of per IPD case was greater than that for OPD, expectedly as this also includes hospitality costs. However, it may be confounded by other morbidities or complications which can prolong patients' time in the hospital and subsequently increases cost. This in turn directly affects the indirect cost since caregivers will spend more days caring for the patient thereby causing productivity losses [35]. The high catastrophic SCD cost observed in this study may not be unrelated to their frequent visit to health facilities and incurring increasing cost, high cost of SCD supplies and late reporting in this sub-region with SCD complications, own money coping payment mechanism and high poverty level in the country. The high incidence of catastrophic costs in this study is worrying because over twenty percent of the households expressed high levels of catastrophic expenditure and the poor is mostly affected. The high indirect cost and its share of catastrophic spending curb off healthcare consumption

because of limited fund for payment. Lack of access to continuing SCD care results in poor health condition and consequently poor quality of life and death. All three of the key preconditions for catastrophic payments identified by Xu et al. [15], were found in this study; the availability of health services requiring payment, low capacity to pay, and lack of prepayment or health insurance. Services are available but there is a high level of out-of-pocket payment which requires payment at the point of care. Over 30% of catastrophic expenditure is very high in a country where more than 70 percent of the population earn less than one US dollar per day and spend more than 40 percent of their total income to satisfy hunger [19]. People paid mostly through out-of-pocket expenditure, with almost no insurance or assured reimbursement payment mechanisms [36].

The findings of the study were subject to several limitations including recall bias associated with the retrospective nature of the survey. The one month recall may have excluded costs incurred quarterly or yearly. Data on premature death and premature retirement were not available and as such were not captured in this study. Also intangible costs such as pain and suffering of SCD patients and families and accompanying reduced quality of life were not include in the analysis. Self-report of cost of SCD given by participants could result in underestimation or exaggeration of the problem.

## Conclusion

Economic burden of SCD was high for the patients and their caregivers. Respondents suffered catastrophic costs.

## Acknowledgments

The authors hereby acknowledge the research assistants: Chidubem, Uju, and Dozie for their commitment and resourcefulness during the data collection and data input.

## Author Contributions

**Conceptualization:** C. N. Amarachukwu, I. L. Okoronkwo.

**Data curation:** C. N. Amarachukwu.

**Formal analysis:** M. C. Nweke.

**Investigation:** C. N. Amarachukwu.

**Methodology:** C. N. Amarachukwu, I. L. Okoronkwo.

**Project administration:** C. N. Amarachukwu.

**Software:** M. C. Nweke.

**Supervision:** I. L. Okoronkwo.

**Validation:** C. N. Amarachukwu.

**Visualization:** M. C. Nweke.

**Writing – original draft:** M. C. Nweke, M. K. Ukwuoma.

**Writing – review & editing:** C. N. Amarachukwu, I. L. Okoronkwo, M. C. Nweke, M. K. Ukwuoma.

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
