## [Decision Letter · Decision Letter 0]

18 Aug 2021

PONE-D-21-17517

Economic burden and catastrophic cost among people living with sickle cell disease, attending a tertiary health institution in south-east zone, Nigeria

PLOS ONE

Dear Dr. Amarachukwu,

Thank you for submitting your manuscript to PLOS ONE. After careful consideration, we feel that it has merit but does not fully meet PLOS ONE’s publication criteria as it currently stands. Therefore, we invite you to submit a revised version of the manuscript that addresses the points raised during the review process.

Major Revision Required.

We look forward to receiving your revised manuscript.

Kind regards,

Ejaz Ahmad Khan, M.D, MPH, FFPH

Academic Editor

PLOS ONE

Journal Requirements:

2. Please include additional information regarding the survey or questionnaire used in the study and ensure that you have provided sufficient details that others could replicate the analyses. For instance, if you developed a questionnaire as part of this study and it is not under a copyright more restrictive than CC-BY, please include a copy, in both the original language and English, as Supporting Information. Moreover, please include more details on how the questionnaire was pre-tested, and whether it was validated.

Reviewers' comments:

Reviewer's Responses to Questions

**Comments to the Author**

1. Is the manuscript technically sound, and do the data support the conclusions?

Reviewer #1: Partly

Reviewer #2: No

2. Has the statistical analysis been performed appropriately and rigorously? 

Reviewer #1: No

Reviewer #2: Yes

3. Have the authors made all data underlying the findings in their manuscript fully available?

Reviewer #1: Yes

Reviewer #2: Yes

4. Is the manuscript presented in an intelligible fashion and written in standard English?

Reviewer #1: Yes

Reviewer #2: No

5. Review Comments to the Author

Reviewer #1: Line 25. "to study a sample of 149 SCD"; Line 146. "the final sample size was 156". Authors should clearly define the sample size.

Line 123. "About 480 SCD patients attended the children clinic and about 400 patients attended the adult clinic in the last one year". How should the term: "in the last one year" be understood?

Line 206. "Total direct cost per patient per month was (US$289.41±316.18), with in-patient direct cost constituting the bulk (US$247.34±260.99)". Is it possible for costs to be negative? The cost written in this way (US$289.41±316.18) I understand as cost form: - 26.77 to 605.59.

Line 216. Table 2. What do IPD and OPD mean?

Reviewer #2: 1.Is the manuscript technically sound, and do the data support the conclusions?

Unfortunately, assigning social status according to the possession of RTV / household equipment is incomprehensible. This is subjective data. Treatment costs are perfectly presented. However, we are not able to estimate the scale of the problem because the earnings of the study group have not been presented. Rtv / household equipment says absolutely nothing about wealth. We have no point of reference. It is difficult for me to compare the cost of treatment in US dollars to the wealth in household appliances, not the amount of earnings in currency. The result is presented that the treatment expenditure is high for patients, and we do not know the patients' income. There is no data on earnings, so we cannot estimate whether the study group was poor or wealthy, nor are we able to say whether treatment is expensive for them or not. Unfortunately.

In addition, data from 6 years ago are presented. This is a very long period and at the moment the situation may (does not have to be) completely different.

Table 1. Quantity and frequency should be split into two columns or the column should be described differently

4. Is the manuscript presented in an intelligible fashion and written in standard English?

Many spaces are missing, especially in the introduction

6. PLOS authors have the option to publish the peer review history of their article (what does this mean?). If published, this will include your full peer review and any attached files.

Reviewer #1: No

Reviewer #2: No

---

## [Author Response · Author response to Decision Letter 0]

1 Dec 2021

The authors wish to thank the editor and reviewers for their comments about this manuscript.

The necessary files have been uploaded and the manuscript arranged according to the journal requirements

Reviewer 1

1. (Line 25 & 146) "To study a sample of 149 SCD"; "The final sample size was 156". Authors should clearly define the sample size

Response: Thank you for your significant contribution. The minimum sample size calculated was 140, but after adjustments for non-response, it became 156. However, 149 participants completed the questionnaire. We have revised the manuscript to capture detailed explanations....see pgs. 2,7,8

2. (Line 123) "About 480 SCD patients attended the children clinic and about 400 patients attended the adult clinic in the last one year". How should the term: "in the last one year" be understood?

Response: September 2014 to October 2015) according to the clinical records....pg. 7

3. (Line 206) "Total direct cost per patient per month was (US$289.41±316.18), with in-patient direct cost constituting the bulk (US$247.34±260.99)". Is it possible for costs to be negative? The cost written in this way (US$289.41±316.18) I understand as cost form: - 26.77 to 605.59.

Response: We subjected the data set to normality test and discovered the data involved were not normally distributed hence we resorted to the statistic of median and corresponding interquartile range. pg. 11

4. (Line 216) Table 2. What do IPD and OPD mean?

Response: The authors thank the reviewer for drawing our attention. We have included the full meanings. pg. 12

Reviewer 2 

Comment 1: Unfortunately, assigning social status according to the possession of RTV/ household equipment is incomprehensible. This is subjective data. Treatment costs are perfectly presented. However, we are not able to estimate the scale of the problem because the earnings of the study group have not been presented. Rtv/ household equipment says absolutely nothing about wealth. We have no point of reference. It is difficult for me to compare the cost of treatment in US dollars to the wealth in household appliances, not the amount of earnings in currency. The result is presented that the treatment expenditure is high for patients, and we do not know the patients' income. There is no data on earnings, so we cannot estimate whether the study group was poor or wealthy, nor are we able to say whether treatment is expensive for them or not. Unfortunately. 

Response: Thank you for your significant contribution and concern. The authors considered the fact that methods of assessing household socio-economic position or status can be categorized into 2 major types: Monetary and Non- monetary. Monetary (Income or expenditure) measures are well understood and more popular. However, criticisms have been observed over using monetary measures to evaluate socio-economic status (SES) especially in low and middle income countries (LMIC) like ours.

These are the challenges of using household income or expenditure as a proxy for classifying SES in LMIC: 

1. Households are often times reluctant to divulge income information and under- reporting of income

2. Difficulty in converting farm products and household gift into income.

3. Measurement errors which are inevitable due to income and expenditure being based on recall memory.

4. Poor quality of income and expenditure in LMIC.

5. Prices of goods, normal interest rates for semi- durable or durable goods are difficult to discern when constructing consumption aggregates.

Given these challenges, proxy indicators have been developed like the wealth index, whereby wealth indices use information about household materials to create index of household wealth. 

Filmer and Pritchett (2001) constructed an asset index by using Principal Component Analysis (PCA).The application of PCA allows researchers to convert series of ownership variables into SES. Collection of asset data have been claimed to be more reliable than income since it uses simple questions or direct observation interview and should therefore suffer less from recall or social desirability bias.

These formed the basis of our using ownership of household equipment other than income as a proxy for wealth assessment. So, the ownership of household equipment was used to classify households into the poorest, the very poor, the poor and the least poor. Table 5 described how catastrophic the spending was on the different socio-economic status as previously grouped.....pg. 7

Comment 2: In addition, data from 6 years ago are presented. This is a very long period and at the moment the situation may (does not have to be) completely different. 

Response: We believe that the data is still valid even after 6 years because all conditions that cause catastrophic spending among people living with sickle cell disease still exists in Nigeria. Out of pocket spending continues to persist as the major mode of payment for healthcare in Nigeria. 

Comment 3: Table 1. Quantity and frequency should be split into two columns or the column should be described differently Response: We have revised accordingly.....pg. 10

Comment 4: Many spaces are missing, especially in the introduction 

Response: We have revised accordingly.....pgs. 3-5

---

## [Decision Letter · Decision Letter 1]

29 Mar 2022

PONE-D-21-17517R1Economic burden and catastrophic cost among people living with sickle cell disease, attending a tertiary health institution in south-east zone, NigeriaPLOS ONE

Dear Dr. %Amarachukwu%,

Thank you for submitting your manuscript to PLOS ONE. After careful consideration, we feel that it has merit but does not fully meet PLOS ONE’s publication criteria as it currently stands. Therefore, we invite you to submit a revised version of the manuscript that addresses the points raised during the review process.

ACADEMIC EDITOR: Please address each comment from each reviewerPlease ensure that your decision is justified on PLOS ONE’s publication criteria and not, for example, on novelty or perceived impact.

We look forward to receiving your revised manuscript.

Kind regards,

Mary Hamer Hodges, MBBS MRCP DSc

Academic Editor

PLOS ONE

Reviewers' comments:

Reviewer's Responses to Questions

**Comments to the Author**

1. If the authors have adequately addressed your comments raised in a previous round of review and you feel that this manuscript is now acceptable for publication, you may indicate that here to bypass the “Comments to the Author” section, enter your conflict of interest statement in the “Confidential to Editor” section, and submit your "Accept" recommendation.

Reviewer #1: All comments have been addressed

Reviewer #2: (No Response)

Reviewer #3: (No Response)

2. Is the manuscript technically sound, and do the data support the conclusions?

Reviewer #1: Yes

Reviewer #2: Partly

Reviewer #3: Partly

3. Has the statistical analysis been performed appropriately and rigorously? 

Reviewer #1: (No Response)

Reviewer #2: N/A

Reviewer #3: I Don't Know

4. Have the authors made all data underlying the findings in their manuscript fully available?

Reviewer #1: (No Response)

Reviewer #2: Yes

Reviewer #3: Yes

5. Is the manuscript presented in an intelligible fashion and written in standard English?

Reviewer #1: (No Response)

Reviewer #2: Yes

Reviewer #3: No

6. Review Comments to the Author

Reviewer #1: (No Response)

Reviewer #2: I still believe that comparing monetary and non-monetary types is not a good idea. Personally, I have no problems with financial liquidity and I would definitely be able to take care of my own and my family's treatment, but according to your classification, you would assign me between the poor and the least poor, which has nothing to do with reality. I am afraid it may be the same in your work. Maybe it was worth using national data, e.g. on the minimum wage in Nigeria. Anything so that a person from different socio-cultural realities could receive hard data and draw their own conclusions. At the moment, it looks like your important work may not be taken seriously because many people, like me, may assign themselves to one of your groups and treat all of your work at a distance.

However, I understand that the point of view may depend on the point of sitting, but think about who is the recipient and what you want to get the message from.

Reviewer #3: This is a difficult paper to understand, and some of the methods need more detailing. Also the presentation of the data needs to be clearer, and the discussion more focused

Major points

I am not sure that the population being studied is representative of the SCD population, as they are attending a tertiary unit, appear to have good electricity supply and >85% have a TV, fridge, and fan, and half have a computer. Of the adults half appear to have post secondary school education.

I assume most SCD patients are seen in primary care if uncomplicated, and then in secondary, with the more severe (and wealthiest) accessing what I assume is more expensive tertiary care. Given the average number of days admission and the high inpatient costs, and frequency of admission in this group, these appear to constitute a more severe phenotype. Generalising from this may not be justified, and certainly merits addressing in the discussion.

The means of determining catastrophic health costs from the proportion of non food expenditure could be more clearly described in the methods (line 198-199). I note the comment and response to the reviewer 2, though see this method as needing greater justification and referencing in the paper. I was surprised that given (line 80) 70% of Nigerians live on <$1 per day the population studied in all quartiles significantly exceeds this.

Minor points

I was not clear on the sample size calculation, and what this was addressing

Patients seem to be having mainly monthly follow up, and longer intervals between visits would be preferred for stable patients- as this would also have an impact on cost.

I assume there is a (local or national) protocol being used- could this be referenced if so. There is also a Nigerian National Guideline for the Control and Management of Sickle Cell Disease 2014, that could be cited if being used. This states:

<<over 300="">In Nigeria, sickle cell disease is among the ten (10) priority non-communicable diseases (NCDs) and it contributes significantly to both child and adult morbidity and mortality. By virtue of its population, Nigeria stands out as the most sickle cell endemic country in Africa with an annual infant death of 100,000 representing 8% of infant mortality in the country. It is also estimated that about 24% Nigerian adults have sickle cell trait.>>

There are two references for the prevalence and morality of SCD in Nigeria, and 2.3% for prevalence seems high. Though the reference given (9) is from a paper on psychological burden. Similarly, the mortality given in line 56 gives mortality as about 5% before age 5. This seems low, and comes from a paper (ref 10) on parental attitude to SCD. Given how few adults there were in the sample, this potentially could indicate high mortality in childhood. Could the authors provide additional references to support these figures?

It would be helpful to know what the costs for standard medications for sickle are in a basic package- and whether newer more expensive treatments were being offered, as the basic package for support is usually not so expensive. And possibly describe what these medicines were, so these OPD costs can be understood (as medicines $15 USD/month). And why physiotherapy was over $20/ month median in OPD.

There are newer treatments for SCD such as hydroxyurea, and in the context of the costs for these patients would be worth considering as a possible means to reduce admissions and may be cost effective in reducing costs associated with more severe disease. The authors may want to provide estimate for the cost of these treatments for an adult per month- though this is above the remit of the paper, it does illustrate the decisions clinicians should have with their patients who have such large OOOP expenditures. Though these in themselves may also put heavy burden on families.

There are a few typos or potential corrections

Lines 174 and 175 – is the low reliability cooeficient of 0.23 in group A or D.

Line 207. I assume ….had SCD diagnosed for one to ten years.

Table 1. Patient age box has 148 patients, as does number of times patients needed care (though n=149). And Variable 4 I think is ‘Age at SCD diagnosis’ rather than ‘Time since diagnosis’

Line 216-218 are repeated. And (line 217) if 28 of the 149 respondents were admitted in the month studied, I struggled to understand how the median costs was $385, as most patients would have only incurred OPD costs- it would be helpful to understand how this small proportion of admissions raised the median cost so much.

Line 253 US$33 should be US$332 (from table 5)

The paper could be improved by significant editing to improve clarity.</over>

7. PLOS authors have the option to publish the peer review history of their article (what does this mean?). If published, this will include your full peer review and any attached files.

Reviewer #1: No

Reviewer #2: No

Reviewer #3: **Yes: **James Bunn

---

## [Author Response · Author response to Decision Letter 1]

7 Jul 2022

We have attached a rebuttal letter addressing reviewers' comments. Thank you

---

## [Decision Letter · Decision Letter 2]

21 Jul 2022

Economic burden and catastrophic cost among people living with sickle cell disease, attending a tertiary health institution in south-east zone, Nigeria

PONE-D-21-17517R2

Dear Dr. %Amarachukwu%,

We’re pleased to inform you that your manuscript has been judged scientifically suitable for publication and will be formally accepted for publication once it meets all outstanding technical requirements.

Kind regards,

Mary Hamer Hodges, MBBS MRCP DSc

Academic Editor

PLOS ONE

Additional Editor Comments (optional):

Thank you for your revisions and patience.

Reviewers' comments:

Reviewer's Responses to Questions

**Comments to the Author**

1. If the authors have adequately addressed your comments raised in a previous round of review and you feel that this manuscript is now acceptable for publication, you may indicate that here to bypass the “Comments to the Author” section, enter your conflict of interest statement in the “Confidential to Editor” section, and submit your "Accept" recommendation.

Reviewer #2: (No Response)

2. Is the manuscript technically sound, and do the data support the conclusions?

Reviewer #2: Yes

3. Has the statistical analysis been performed appropriately and rigorously? 

Reviewer #2: Yes

4. Have the authors made all data underlying the findings in their manuscript fully available?

Reviewer #2: Yes

5. Is the manuscript presented in an intelligible fashion and written in standard English?

Reviewer #2: Yes

6. Review Comments to the Author

Reviewer #2: (No Response)

7. PLOS authors have the option to publish the peer review history of their article (what does this mean?). If published, this will include your full peer review and any attached files.

Reviewer #2: No

---

## [Editor Report · Acceptance letter]

2 Aug 2022

PONE-D-21-17517R2 

Economic burden and catastrophic cost among people living with sickle cell disease, attending a tertiary health institution in south-east zone, Nigeria. 

Dear Dr. Amarachukwu:

I'm pleased to inform you that your manuscript has been deemed suitable for publication in PLOS ONE. Congratulations! Your manuscript is now with our production department. 

Kind regards, 

on behalf of

Prof. Mary Hamer Hodges 

Academic Editor

PLOS ONE